# Profiling of Seed Proteome in Pea (*Pisum sativum* L.) Lines Characterized with High and Low Responsivity to Combined Inoculation with Nodule Bacteria and Arbuscular Mycorrhizal Fungi

**DOI:** 10.3390/molecules24081603

**Published:** 2019-04-23

**Authors:** Tatiana Mamontova, Alexey M. Afonin, Christian Ihling, Alena Soboleva, Elena Lukasheva, Anton S. Sulima, Oksana Y. Shtark, Gulnara A. Akhtemova, Maria N. Povydysh, Andrea Sinz, Andrej Frolov, Vladimir A. Zhukov, Igor A. Tikhonovich

**Affiliations:** 1Department of Biochemistry, St. Petersburg State University, 199178 St. Petersburg, Russia; mamontova-bio@mail.ru (T.M.); oriselle@yandex.ru (A.S.); elena_lukasheva@mail.ru (E.L.); 2Department of Bioorganic Chemistry, Leibniz Institute of Plant Biochemistry, 06120 Halle (Saale), Germany; 3Department of Biotechnology, All-Russia Research Institute for Agricultural Microbiology, 196608 St. Petersburg, Russia; afoninalexeym@gmail.com (A.M.A.); asulima@arriam.ru (A.S.S.); oshtark@yandex.ru (O.Y.S.); ahgulya@yandex.ru (G.A.A.); vladimir.zhukoff@gmail.com (V.A.Z.); 4Department of Pharmaceutical Chemistry and Bioanalytics, Institute of Pharmacy, Martin-Luther Universität Halle-Wittenberg, 06120 Halle (Saale), Germany; christian.ihling@pharmazie.uni-halle.de (C.I.); andrea.sinz@pharmazie.uni-halle.de (A.S.); 5R&D Department, Saint-Petersburg State Chemical and Pharmaceutical University, 197376 St. Petersburg, Russia; maria.povydysh@pharminnotech.com; 6Department of Genetics and Biotechnology, St. Petersburg State University, 199034 St. Petersburg, Russia

**Keywords:** arbuscular mycorrhiza, beneficial soil microorganisms, combined inoculation, inoculation efficiency, LC-MS, pea (*Pisum sativum* L.), proteomics, rhizobial symbiosis, seed proteome

## Abstract

Legume crops represent the major source of food protein and contribute to human nutrition and animal feeding. An essential improvement of their productivity can be achieved by symbiosis with beneficial soil microorganisms—rhizobia (Rh) and arbuscular mycorrhizal (AM) fungi. The efficiency of these interactions depends on plant genotype. Recently, we have shown that, after simultaneous inoculation with Rh and AM, the productivity gain of pea (*Pisum sativum* L) line K-8274, characterized by high efficiency of interaction with soil microorganisms (EIBSM), was higher in comparison to a low-EIBSM line K-3358. However, the molecular mechanisms behind this effect are still uncharacterized. Therefore, here, we address the alterations in pea seed proteome, underlying the symbiosis-related productivity gain, and identify 111 differentially expressed proteins in the two lines. The high-EIBSM line K-8274 responded to inoculation by prolongation of seed maturation, manifested by up-regulation of proteins involved in cellular respiration, protein biosynthesis, and down-regulation of late-embryogenesis abundant (LEA) proteins. In contrast, the low-EIBSM line K-3358 demonstrated lower levels of the proteins, related to cell metabolism. Thus, we propose that the EIBSM trait is linked to prolongation of seed filling that needs to be taken into account in pulse crop breeding programs. The raw data have been deposited to the ProteomeXchange with identifier PXD013479.

## 1. Introduction

Due to a variety of interactions with a wide range of beneficial soil microorganisms (BSM), legumes attract a special interest of microbiologists, molecular biologists, and plant biochemists [1,2,3]. Indeed, these plants can be involved in at least three forms of mutualistic plant–microbial interactions (symbioses): (i) with Glomeromycota fungi in arbuscular mycorrhizae (AM) [4], (ii) with nitrogen-fixing bacteria (rhizobia) in root nodules (legume–rhizobial symbiosis, LRS) [5], and (iii) with plant growth promoting bacteria (PGPB) [6]. The biological roles of these associations are clearly different. AM facilitates the assimilation of sparingly soluble phosphates by plant roots and increases efficiency of water consumption [4], whereas LRS acts as the basis for biological fixation of atmospheric nitrogen [3]. Interaction with PGPB additionally provides defense against pathogenic microorganisms suppressing their growth [7,8]. Moreover, synergistic effects of colonization with different BSM on legume plants are well characterized. Indeed, triple inoculation (i.e., simultaneous colonization with AM fungi, rhizobia, and PGPB) results in higher biomass and seed yields, compared to inoculation with only one or two BSM [1,9,10,11,12].

In general, formation of AM improves soil structure, whereas LRS results in accumulation of bioavailable nitrogen in soil [13]. Hence, these symbioses are beneficial for the whole rhizosphere [10,13]. Moreover, at the ecosystem level, legume–AM–rhizobial symbiosis may impact seedling development, plant biodiversity, and nutrition [14]. Therefore, cultivation of legumes can lead to reduction in use of mineral fertilizers and pesticides in favor of biological agents (i.e., these crops are ideal for sustainable agriculture) [15,16], making it possible to maintain their productivity under unfavorable conditions [9]. 

In general, the exact molecular mechanisms underlying the regulation of BSM-related symbiosis are still poorly understood, although the involvement of over 50 legume genes was confirmed (mostly encoding key players of signal transduction pathways and metabolic regulatory networks) [2]. On another hand, establishment of LRS [17] and AM [18] is accompanied by alterations in expression of thousands of genes, hundreds of which are represented by the so-called symbiosins [19]. Moreover, diverse transcriptional responses (known as specific genome-wide signatures) are associated with synergistic benefits of multiple interconnected mutualistic associations, as was shown, for example, for a tripartite association of *Medicago truncatula*, rhizobia, and AM fungi [20]. These changes in transcription patterns, in turn, directly affect proteome and metabolome profiles [21]. Thereby, due to the integrative character of plant regulatory systems, not only roots (as the organs, directly involved in interaction with bacteria and fungi), but also various parts of shoots, can response to inoculation with symbiotic organisms. Obviously, in the case of crop plants, it might affect commercial properties of plant-derived foods, which, for legumes, are mostly seeds. Due to the high importance of legume seeds for world production of food protein [22], symbiosis-related changes in seed proteome need to be comprehensively characterized. In this context, a bottom-up proteomic approach is a powerful tool to address both dynamics of individual proteins and patterns of post-translational modifications [23,24], potentially harmful to humans [25]. Certainly, these changes, related to root symbiosis, need to be considered in the context of plant biomass gain and seed productivity [24].

Pea (*Pisum sativum* L) is a wide-spread crop plant, highly variable in terms of efficiency of interaction with nodule bacteria *Rhizobium leguminosarum* and AM fungus *Rhizophagus irregularis* [26,27]. This variability, usually referred to as “efficiency of interaction with BSM” (EIBSM) can only be studied with complex BSM inoculants containing rhizobia and a combination of several AM fungi [1,28]. In a wide-scale study with 26 pea genotypes, a high efficiency of combined BSM inoculates in respect to biomass accumulation and seed protein contents was confirmed [29].

In this study we compare seed proteome profiles of the pea lines K-8274 (cv. Vendevil, France) and K-3358 (local landrace from Saratov region, Russia) [27], characterized with high and low EIBSM, respectively. The plants of both lines were grown in presence and absence of combined BSM. Thereby, inoculants contained *R. leguminosarum* and a combination of three different *R. irregularis* isolates. Comparison of these contrasting lines allowed to comprehensively characterize the changes in the proteomes of mature seeds related to complex mutual symbiosis. It also allowed uncovering the influence of the symbiotic efficiency trait on the development of seeds of different lines and identification the proteins likely responsible for the exhibited differences between the investigated lines. 

## 2. Results

### 2.1. Biomass Gain and Seed Productivity

Previously, in three-year field trials, the line K-8274 has been chosen as a “standard” for high EIBSM, since it demonstrated a significant and stable increase in shoot and seed biomass upon the complex inoculation with AM fungi and nodule bacteria [29]. Interestingly, inoculation of the K-8274 plants with individual rhizobial or AM-fungal cultures did not result in increase of shoot or seed biomass production (Appendix A) [30]. On another hand, simultaneous root colonization of the K-8274 plants with the same two BSM (i.e., rhizobia and AM fungi) resulted in significant gain in shoot and seed biomass (t-test: *p* = 0.02 and 0.004, respectively, Appendix A). Based on this fact, we decided to investigate the phenomenon of high EIBSM in more detail, and addressed the alterations in seed proteome underlying high responsivity to combined inoculation with rhizobia and AM-fungi. For this, we reproduced the combined inoculation setup in a pot experiment with the high-EIBSM K-8274 plants in parallel to the line K-3358, characterized with low EIBSM [29]. Although the low EIBSM line K-3358 was characterized with a 23% and 25% higher degree (in comparison to the line K-8274) of shoot and seed biomass accumulation, respectively, its inoculation with the combination of two BSM did not give additional gain in productivity (Appendix A). 

### 2.2. Protein Isolation and Tryptic Digestion

To achieve quantitative isolation of seed proteins, and the maximal coverage of a mature seed proteome, we decided for the phenol extraction procedure (Figure 1). The minimal concentration of the anionic acid-labile surfactant (AALS) required for quantitative solubilization of the protein was 0.15% (*w*/*v*), so it was used herein. According to the results of the Bradford assay, yields of the protein extraction were in the range of 63.6–175.0 mg/g fresh weight (Appendix A). This was confirmed by the sodium dodecyl sulfate-polyacrylamide gel electrophoresis (SDS-PAGE) analysis, performed with the sample load, calculated based on the results of the Bradford assay [31]; the whole lane average intensities obtained with equal sample amounts (5 µg of protein) were 1.77 × 10^4^ ± 2.25 × 10^3^ (RSD = 12.7%). Thereby, the patterns of the signals, observed in the electropherograms were similar between lanes and treatment groups. The subsequent tryptic digestion (Figure 1) was considered to be complete, as the bands of major pea storage proteins, as legumin (α- and β- subunits, ∼40 and ∼20 kDa, correspondingly), vicilin (subunits of ∼29, ∼35, and ∼47 kDa) and convicilin (subunit of ∼71 kDa), were not detectable [31], assuming a staining sensitivity better than 30 ng [32] and a legumin content of at least 80% of the total seed protein [33].

### 2.3. Identification of Seed Proteins

Selection of an appropriate sequence database is the pre-requisite for successful identification of proteolytic peptides in enzymatic digests and, hence, reliable annotation of seed proteins. In this context, the use of reviewed databases, containing entries confirmed at the level of transcriptome or proteome, is advantageous. However, such information is not readily available for pea. Therefore, here we decided on a non-redundant combined database relying on several legume proteomes, closely related to pea—*Medicago truncatula* Gaertn, *Lotus japonicas* (Regel) K. Larsen, and *Phaseolus vulgaris* L. Earlier, to confirm the applicability of this database, we manually evaluated the MS/MS spectra of confidently identified peptides with the lowest values of the SEQUEST function XCorr [34]. As the spectra were acquired with the mass accuracy within 5 ppm, peptide sequences could be unambiguously assigned by characteristic patterns of N- and C-terminal ion series ( b and y ions, respectively) [31].

Analysis of the seed proteome of both lines resulted in confident identification of 3963 peptides in total (3557 and 3726 in the seeds of K-8274 and K-3358, respectively, Figure 2A, Appendix A). Based on this information, 5832 proteins were annotated (5195 in the seeds of K-8274 and 5593 in K-3358, Figure 2B), which represented 1500 non-redundant proteins (i.e., protein groups—1346 and 1425 in the seeds of K-8274 and K-3358, respectively, Figure 2C). For the line K-8274, 84 non-redundant proteins could be annotated only in the absence of BSM, whereas 103 features were found specifically in the seeds of inoculated plants. For K-3358, these values were 114 and 69, respectively (Figure 2C). The numbers of non-redundant proteins, not dependent on inoculation, were 1159 for the line K-8274 and 1242 for the line K-3358, with 1101 being, overall, common for both lines. Interestingly, only 12 such proteins were common between the seeds of both lines in combined symbiosis with rhizobia and AM fungi, and 15 proteins were common between the seeds of plants grown without BSM.

### 2.4. Label-Free Quantification

Analysis of the whole dataset with the Progenesis QIP software revealed 79 differentially expressed proteins (ANOVA, *p* ≤ 0.05). Additionally, a further 32 proteins were identified with the original redundant database, containing non-reviewed entries. The correctness of these identifications was confirmed by manual interpretation of the corresponding MS/MS spectra (Appendix A). Thus, 111 proteins were differentially expressed (as could be proved by verification of peak integration, Appendix A), in other words, demonstrated at least 1.5-fold significant abundance differences in intra- and inter-line comparisons (Table 1, Appendix A). One of the raw files (corresponding to one of the triplicates of not inoculated group of line K-8274) could not be satisfactory aligned to the whole dataset, and was therefore excluded from quantitative analysis.

Among regulated seed proteins, 84 were differentially expressed between the lines in the inoculated (BSM) and 99 in the not inoculated (NI) group. Remarkably, 36 and 61 proteins were more abundant in the BSM and NI groups of K-3358 plants, respectively, in comparison to the same groups of the K-8274 line. In contrast, the abundance of 48 and 38 proteins in BSM and NI groups of K-3358 plants was lower in comparison to K-8274 plants. Totally, 60 proteins in the seeds of K-8274 demonstrated inoculation-related changes in expression profiles (50 and 10 polypeptides were up- and down-regulated upon combined inoculation with BSM, respectively). For the line K-3358, these values were 31 and 29, respectively. 

Principle component analysis (PCA) revealed clear differences between two lines, which could be distinguished by the first component (67.2% of difference, Figure 3A,B and Appendix A corresponding loading plots are given on Appendix A). For each line, the differences between inoculated and not inoculated plants were much less pronounced, although clearly observable (3.3% and 7.6% differences in the components 2 and 3, respectively, Figure 3A,B, respectively). At the next step, hierarchical clustering was applied to classify individual differentially regulated proteins according to their intra- and inter-line differences in expression profiles. Based on the heat map, built for average values of each group (Figure 3C), all differentially expressed non-redundant proteins could be assigned to one of 17 individual groups, organized by similarity of expression profiles (Appendix A). The original results of data clustering in Perseus are given on Appendix A. Finally, depending on the direction of protein expression changes, these groups (further referred to as sub-clusters) were organized in ten principle clusters. Thus, response of individual proteins to inoculation with combined BSM, could be expressed as “up-regulated”, “down-regulated”, and “not responsive” or “steady” relative to corresponding NI controls. Thus, combination of these regulation states in two lines yielded nine principle clusters (i.e., clusters 1–3, 4–6, and 7–9) comprising the proteins up-regulated, down-regulated, and not responsive in line K-8274, respectively, with different regulation status of the line K-3358 (Table 1). The last cluster (#10) comprised the proteins, identified only in one of the lines (Table 1).

The first principle cluster represented non-redundant proteins, abundance of which increased in the seeds of both lines in response to combined inoculation with BSM. Similarly, the fifth cluster represented the proteins with decreased abundances (i.e., down regulated) in both lines in response to the inoculation, whereas the non-responsive proteins built the ninth cluster. The proteins comprising the second cluster were up-regulated in the line K-8274, but down-regulated within line K-3358. This principle cluster consisted of two sub-clusters: The proteins, demonstrating the lowest abundance in (i) NI group of line K-8274, and (ii) in the BSM group of line K-3358. The proteins of the forth cluster demonstrated inverse response to inoculation. This principle cluster also consisted of two sub-clusters: Demonstrating the lowest abundance (i) in BSM group of line K-8274 and (ii) in the NI group of line K-3358. The next group of principle clusters was represented by the seed proteins, regulated by inoculation with BSM only in one of the lines. Thus, the proteins of the clusters 3 and 6 (both including two sub-clusters) were up- and down-regulated in the seeds of K-8274, respectively, but demonstrated a “steady” behavior in the line K-3358. Analogously, the proteins of the clusters 7 and 8 (represented by one and three sub-clusters, respectively) were up- and down-regulated in response to inoculation with BSM in the seeds of line K-3358, respectively, with no abundance changes in the seeds of the line K-8274. Finally, β-hexosaminidase was found only in the seeds of line K-3358, and probable S.7-like l-type lectin-domain containing receptor kinase was identified in the line K-8274. These two proteins comprised the last, tenth cluster.

### 2.5. Functional Annotation of Differentially Regulated Proteins

Functional annotation of differentially expressed proteins relied on the Mercator tool, and revealed clear inter-line differences in functional profiles of regulated proteome (Appendix A). Forty of the 60 proteins, changing their abundance in the seeds of the high-EIBSM line K-8274 upon combined inoculation, were successfully assigned to specific functional bins. Totally, 50 of the 60 proteins (including 34 assigned to functional bins) were up-regulated (Figure 4A). Protein biosynthesis represented the most strongly affected function—only one polypeptide, namely 60S ribosomal protein L26-1, was down-regulated. Such processes as RNA biosynthesis, RNA processing, protein modification, and degradation were affected as well. Accordingly, symbiosis-related up-regulation of energy metabolism was observed: Three enzymes of cellular respiration and two enzymes involved in photosynthesis increased their abundance in response to colonization of K-8274 roots. Another strongly up-regulated function was chromatin organization—three different types of core histones increased their abundance, which was in line with the overall up-regulation of RNA and protein biosynthesis. 

In agreement with the fact that approximately half of differentially expressed proteins were up-regulated in K-3358 seeds in response to interaction with BSM (31 of 60), the number of annotated up-regulated proteins was 17 out of overall 36 successfully annotated species (Figure 4B). Remarkably, in the seeds of the K-3358 plants, the proteins, involved in protein biosynthesis, showed more prominent difference in expression profiles: Besides the seven up-regulated polypeptides, four species, namely UTP-glucose-1-phosphate uridylyltransferase, 60S ribosomal protein L3B, and two poorly characterized probable structural constituents of ribosome, decreased their abundance after inoculation with BSM. The large number of polypeptides involved in protein biosynthesis, including ribosomal proteins and the EF2 elongation factor, were up-regulated in both lines upon inoculation with BSM. Interestingly, the exact set of ribosomal proteins, up-regulated in presence of symbiosis with BSM, was different in two lines, possibly reflecting the difference in response of the microorganisms to the symbiosis. The same was the case only for the proteins involved in cellular respiration (e.g., ATP synthase subunit alpha and probable triosephosphate isomerase), which were clearly up-regulated. The triosephosphate isomerase was shown to be required for post-germinative transition to autotrophic growth in seeds [36]. The significant increase of ATP-synthase abundance points at the significant increase in metabolism of the seeds of the line K-8274 upon symbiosis. 

Remarkably, in the K-3358 line, several functional protein groups were exclusively down-regulated upon inoculation with BSM. This can be exemplified by the proteins involved in redox homeostasis (catalase, probable peroxiredoxin (UniProt ID B7FH22), and superoxide dismutase), protein degradation and modification (proteasome alpha subunit, serine carboxypeptidase-like protein, and glutathione S-transferase), and vesicle tracking (clathrin heavy chain and GDP-dissociation inhibitor). All these observations indicate a decrease in metabolic activity in the seeds of the low-EIBSM line in comparison to those of the high-EIBSM one. Interestingly, probable peroxiredoxin (B7FH22) and phosphoenolpyruvate carboxylase, involved in redox homeostasis and photosynthesis are down-regulated in this line, while they are among the up-regulated species in the seeds of the line K-8274 (Figure 4). Among the proteins without assigned functional category, a polypeptide annotated as an embryogenesis abundant protein, significantly decreased its abundance in seeds of K-8274 and increased it in K-3358. Proteins of this group were mostly found in seeds at late developmental stages [37] and can thus be related to seed maturity. 

Prediction of cellular localization, performed for differentially expressed proteins, revealed cytosol as the major cellular fraction, responding to inoculation with BSM, although nuclear and plastid proteins were highly represented as well (Figure 5, Appendix A). Thus, plastid proteins constituted the most symmetrically changing group of proteins; in both lines, these species represented 10%–12% of up- and down-regulated polypeptides. On the other hand, the most variable groups were represented by extracellular (up to 20% of all of the proteins), membrane (up to 10%), and mitochondrial (up to 10%) proteins (Figure 5, Appendix A). Interestingly, vacuolar proteins represented the only down-regulated localization protein group within both lines (Figure 5B,D). Some proteins were identified exclusively in specific groups: One protein with shared localization in Golgi apparatus and mitochondrion was up-regulated within line K-8274, while two peroxisome proteins were found only down-regulated in line K-3358 (Figure 5A,D, respectively), also indicating a differential response of metabolism of the seeds to inoculation.

## 3. Discussion

### 3.1. Complex Inoculation Affects Seed Productivity only in the High-EIBSM Line K-8274

During the recent years, legume seed proteome was intensively studied [34,38]. Specifically, Sistani et al. addressed the changes in pea seed protein content upon inoculation with rhizobia and/or arbuscular-mycorrhizal fungi in the context of resistance to the pathogenic fungi *Didymella pinodes* [39]. Here we consider these inoculation-related effects with a specific focus to susceptibility of plants to inoculation with BSM. For this, we employ two pea lines with contrasting EIBSM as an efficient tool to dissect metabolic effects underlying the observed increase in seed protein contents. In agreement with this aim, we rely here on the methods of bottom-up proteomics—an efficient functional genomics tool, well-established in seed research during the last decade [40]. Recently, we validated our nanoHPLC-ion trap (IT)-Orbitrap-MS-based approach for label-free quantification and confirmed its high reliability, precision, and sensitivity [32]. In our earlier studies it proved to be well compatible with other functional genomics techniques [24]. 

At the level of morphology, the beneficial effect of complex inoculation (namely, increase in seed/shoot biomass and seed number) was observed for the high-EIBSM line K-8274, but not the low-EIBSM one K-3358, which was in agreement with the previous studies [41]. Thereby, as inoculation of K-8274 with individual BSM did not result in any significant beneficial effects on plant productivity [30], only the effects of complex inoculation (with rhizobia and AM fungi simultaneously) were addressed here. It is well known, that individual pea lines strongly differ in their response to inoculation with individual BSM and their combinations. Indeed, for the pea cultivar Messire, double inoculation was inefficient [42], whereas root colonization with an individual culture of rhizobia resulted in significant gain in seed weight [39]. This example clearly illustrates the importance of using contrasting genotypes for these kinds of studies. 

### 3.2. Differences in Protein Expression Patterns between the Pea Lines with High and Low EIBSM

The overall success of comprehensive proteome characterization critically depends on proper protein identification methods. Therefore, here we applied a representative sequence database of legume species, closely related to pea. This approach proved to be efficient in our previous studies [44]. Thus, altogether 1500 non-redundant proteins were identified here (1346 and 1425 in the seeds of K-8274 and K-3358, respectively). Although it was slightly lower in comparison to the results of our recent comprehensive profiling of pea seed proteome [34], the conclusions drawn here are still based on the most complete, to the best of our knowledge, pea seed protein map.

In agreement with the results of Turetschek et al. [45], the effect of plant genotype on seed proteome signatures was more pronounced, than the impact of inoculation with BSM. Thus, due to the contrasting seed color (green and yellow for K-3358 and K-8274, respectively), several proteins related to photosynthesis were differently expressed in two analyzed lines. Indeed, the yellow color of seeds of the K-8274 plants was due to the lack of the active SGR (STAY GREEN) protein, involved in regulation of chlorophyll degradation, encoded by the gene I [46,47]. Further, the proteins involved in abscisic acid (ABA) signaling were clearly more abundant in the seeds of K-8274 in comparison to the seeds of the low-EIBSM line. The role of these molecules (17.6 kDa class I heat shock protein, translation elongation factor EF-2 subunit, UTP-glucose-1-phosphate uridylyltransferase, and ABA-responsive protein, Table 1, Appendix A) in ABA signal transduction is well-characterized in non-legume species [48,49,50,51]. As ABA is a critical regulator of late steps of seed development [47], this observation might indicate inter-line differences in seed maturation rates. We also identified proteins differentially expressed in two lines that might indicate high polymorphism of pea seeds in respect of their proteome signatures. Thus, the approach relying on relative quantification of individual proteins might have a high value in breeding. This conclusion is supported by the work of Bourgeois et al. [52], where the genetic architecture of seed proteome variability was uncovered and the protein quantity loci, responsible for different seed protein composition and protein content, were identified.

### 3.3. Response of High- and Low-EIBSM Pea Lines to Inoculation with Rhizobia and Arbuscular Mycorrhiza

In general, the observed responses of seed proteome to inoculation with BSM could be classified as line-unspecific and line-specific. The non-specific responses manifested as up-regulation of the polypeptides, involved in protein biosynthesis and vesicle transport. This fact might indicate an improved availability of soil phosphorous and nitrogen. However, the number of such hits was lower in comparison to the proteins, demonstrating inter-line expression differences (as shown on the PCA plots on Figure 4A,B; corresponding loading plots on Appendix A). As these proteins could contribute on the observed difference in EIBSM, we addressed this group in more detail.

The analyzed lines showed a differential response to inoculation with BSM. Thus, K-8274 demonstrated stronger symbiosis-related differences in expression of seed proteins in comparison to the low-EIBSM line. Moreover, the functional patterns of the expression differences were clearly line-specific. Thus, inoculation of the low-EIBSM line K-3358 with two BSM resulted in down-regulation of the proteins involved in central and energy metabolism, as well as biosynthesis and post-translational modification of proteins. In agreement with this, the K-3358 plants inoculated with BSM completed seed development earlier, than corresponding not inoculated controls.

In contrast, inoculation of the high-EIBSM line K-8274 resulted in up-regulation of the polypeptides, involved in biosynthetic pathways, cellular respiration, detoxification of reactive oxygen species (ROS), and photosynthesis. One of the up-regulated biosynthetic enzymes, namely phosphoenolpyruvate carboxylase, was previously shown to be highly correlated with seed protein and lipid contents in soybean [53]. On another hand, a plastid protein with triose phosphate isomerase activity appeared to be up-regulated (Appendix A). A protein with this activity was earlier shown to be crucial for post-germinative switch from heterotrophic to autotrophic growth in Arabidopsis [36]. The observed inoculation-related changes might indicate a high level of cell metabolism, which is essential for seed filling and beneficial for seed development. Differential expression of some other proteins might be related to line-specific differences in interaction of pea plants with symbiotic bacteria. Thus, β-hexosaminidase, expressed exclusively in K-3358 seeds, was not earlier reported in the context of legume–rhizobial symbiosis. On the other hand, the S.7-like l-type lectin-domain containing receptor kinase, found only in the K-8274 seeds, can potentially be involved in the reception of microorganisms and thus potentially may represent a link between the inoculation and seed formation. 

In agreement with this, the high-EIBSM line K-8274 demonstrated an inoculation-related down-regulation of late embryogenesis abundant (LEA) protein A0A072TMR3. This might indicate retardation of seed maturation. Indeed, a similar observation was done by Sistany et al. [39], who reported lower levels of LEA proteins in pea plants, inoculated with rhizobia, in comparison to corresponding non-inoculated controls. Thus, we assume that the high-EIBSM genotype of the K-8274 line might contribute to the prolongation of the immature stage of seed development upon the inoculation with BSM, whereas the low-EIBSM line K-3358 did not respond to complex symbiosis in this way. Recently, we have shown that arbuscular mycorrhiza results in prolongation of the pea life cycle (Shtark et al., under revision). Therefore, we assume the mycorrhizal component of the inoculum to be the main contributor to the inoculation-related seed biomass increase, observed in this study for the high-EIBSM K-8274 line.

Another important marker of the inoculation-related retardation in seed maturation is 1,2-dihydroxy-3-keto-5-methylthiopentene dioxygenase—an enzyme involved in methionine salvage and annotated here by the *M. truncatula* part of our combined sequence database as MEDTR1G102870.1. According to the *M. truncatula* gene expression atlas [54], this enzyme is expressed in most of the tissues. Thereby, in seeds, it shows a characteristic expression pattern (i.e., its abundance decreases from the 10th to the 24th day after pollination (DAP), and increases until the 36th DAP. Under symbiotic conditions, this increase in abundance was four-fold in the seeds of K-8274, whereas this enzyme was two-fold down-regulated in the seeds of K-3358 plants, inoculated with BSM. This observation was in agreement with the here proposed prolongation of the immature stage of seed development in the high-EIBSM line upon the inoculation with BSM. As MEDTR1G102870.1 can be a promising marker of this “prolonged seed youth” phenomenon, expression levels of the corresponding gene and kinetics of the enzymatic reaction product deserve to be determined in future studies.

### 3.4. Ecological and Agricultural Aspects of the High-EIBSM Trait 

Most probably, the differential response to inoculation with BSM (i.e., low- and high-EIBSM traits) reflects two strategies of nitrogen assimilation upon its supplementation: Some genotypes demonstrate prolonged seed filling under optimal nitrogen supply conditions, whereas the others complete seed development as fast as possible (reminiscent to r- and K-strategies characteristic for different higher organisms [55]). Obviously, representation of the both strategies in a population might increase its overall adaptation flexibility. On another hand, the difference in response to available nitrogen can be attributed to the breeding history of individual pea varieties and cultivars. In this context, we assume that the plants of the low-EIBSM line prioritize the speed of seed maturation over the maximization of the nutrient content of the seeds. Interestingly, the K-3358 plants develop multiple reproductive nodes (i.e., new seeds can form during the whole ontogenesis). In contrast, the K-8274 plants can produce only a limited number of reproductive nodes, and, hence, develop a limited pre-determined number of pods and seeds, in which the available resources invested. Thus, K- and r-strategies of different pea genotypes might reflect corresponding growth patterns. Most probably, at the metabolic level, these strategies can be due to differences in (i) nitrogen sensing, (ii) efficiency of nitrogen uptake from soil or efficiency of its fixation in nodules, and (iii) assimilation of nitrogen by the seeds.

Seed development is, metabolically, closely associated with re-mobilization of nitrogen from vegetative tissues to seeds, which triggers leaf senescence and shortens seed filling period [56]. On another hand, mycorrhization prolongs the metabolically active stages of leaf ontogenesis (i.e., Shtark et al., under revision). Thus, high-EIBSM genotypes, like K-8274, represent well-balanced systems with improved efficiency of seed filling due to a longer immature stage in seed development. Therefore, such genotypes give access to higher biomass gain. Hence, involvement of high-EIBSM lines in breeding programs might increase the overall agricultural efficiency. One needs to keep in mind; however, that environmental stress, like drought, common in most pea culturing countries, might eliminate the favorable effects of symbiosis with BSM [57]. Therefore, additional experiments in adequate drought models [58] are necessary to address inoculation of high-EIBSM pea plants with BSM under conditions of environmental stress.

## 4. Materials and Methods 

### 4.1. Reagents

Unless stated otherwise, materials were obtained from the following manufacturers. Carl Roth GmbH and Co (Karlsruhe, Germany): acetonitrile (≥99.95%, LC-MS grade), ethanol (≥99.8%), sodium dodecyl sulfate (SDS) (>99%), tris-(2-carboxyethyl)-phosphine hydrochloride (TCEP, ≥98%); PanReac AppliChem (Darmstadt, Germany): acrylamide (2K Standard Grade), glycerol (ACS grade); AMRESCO LLC (Fountain Parkway Solon, OH, USA): ammonium persulfate (ACS grade), glycine (biotechnology grade), *N*,*N*′-methylene-bis-acrylamide (ultra-pure grade), tris(hydroxymethyl)aminomethane (tris, ultra-pure grade), urea (ultra-pure grade); Bioanalytical Technologies 3M Company (St. Paul, MN, USA): Empore™ solid phase octadecyl extraction discs; Component-Reactiv (Moscow, Russia): phosphoric acid (p.a.); Reachem (Moscow, Russia): hydrochloric acid (p.a.), isopropanol (reagent grade), potassium chloride (reagent grade); SERVA Electrophoresis GmbH (Heidelberg, Germany): Coomassie Brilliant Blue G-250, 2-mercaptoethanol (research grade), trypsin NB (sequencing grade, modified from porcine pancreas); Thermo Scientific (Waltham, MA, USA): PierceTM Unstained Protein Molecular Weight Marker #26610 (14.4–116.0kDa); Dichrom GmbH (Marl, Germany): Progenta™ anionic acid labile surfactant II (AALS) and adaptors for stage-tips; Vekton (Saint-Petersburg, Russia): sucrose (ACS grade). All other chemicals were purchased from Sigma-Aldrich Chemie GmbH (Taufkirchen, Germany). Water was purified in house (resistance 5–15 mΩ/cm) on a water conditioning and purification system «Elix 3 UV» (Millipore, Moscow, Russia). The seeds of pea (Pisum sativum L) lines with accession numbers K-8274 (cultivar Vendevil, France) and K-3358 (local landrace from Saratov region, Russia), characterized by high and low EIBSM, respectively, were initially obtained from the collection of the Vavilov Institute of Plant Genetic Resources (St. Petersburg, Russia) and were propagated prior to the experiment in ARRIAM (St. Petersburg, Russia).

### 4.2. AM Fungal Inoculum

The AM inoculum relied on a combination of three *R. irregularis* strains, namely BEG144, BEG53 (both provided by the International Bank for the Glomeromycota, Dijon, France), and ST3 (All-Russia Research Institute for Agricultural Microbiology, Saint-Petersburg) [59]. All isolates were cultured individually in a sand/soil mixture (1:1 *v*/*v*) using *Plectranthus australis* R. Br. as a host plant. To obtain the inoculum of AM fungi, the seeds of sorghum (*Sorghum sp*.) were surface sterilized with a 0.15% (*w*/*v*) aqueous solution of potassium permanganate for 15 min, and transferred to pots, filled with a soil-based substrate (pH 7) containing dried *P. australis* roots colonized with the three above mentioned *R. irregularis* strains. After about 120 days of vegetation, the colonized sorghum roots were separated from the substrate, cut into 1 cm pieces, dried and mixed with the substrate to establish the inoculum. 

### 4.3. Plant Experiments and Characterization of Biomass Gain and Seed Productivity

The seeds were surface sterilized with concentrated sulfuric acid, rinsed with sterile water, germinated on wet vermiculite for three days in darkness at 25 °C, planted in 5 L pots filled with sod-podzolic light loamy soil (five plants per pot), and inoculated with 150 ml of water suspension (106 CFU * l-1) of symbiotic bacteria (*Rhizobium leguminosarum* bv. *viciae* RCAM1026) [60] in combination with prepared inoculum (see previous section). Thereby, the planted seeds (*n* = 5) were overlaid with 30 g/pot of the AM fungal inoculum (see previous section). Before planting, the weight of pots was adjusted with soil to obtain the same value. The plants were grown under non-controlled light and temperature conditions in a vegetation house of the All-Russia Research Institute for Agricultural Microbiology, St. Petersburg (June–August 2016). Formation of AM was verified on the 28th day after germination by light microscopy, as described by Shtark et al. [61]. The plants were harvested at the stage of mature seeds (3 months after planting), and the dry weight of aerial part, the weight of seeds and the total number of seeds per plant, were recorded. Data processing and statistical evaluation was done with SigmaPlot 12.0 software (Systat Software, San Jose, CA, USA).

### 4.4. Protein Isolation 

Pea seeds (10 per biological replicate) were frozen in liquid nitrogen and ground in a Mixer Mill MM 400 ball mill with a Ø 20 mm stainless steel ball (Retsch, Haan, Germany) at a vibration frequency of 30 Hz for 2 × 1 min, and kept on dry ice prior to protein extraction. The total protein fraction was isolated from the frozen ground material by phenol extraction, as described by Frolov and co-workers [62] with some modifications. Briefly, approximately 50 mg of plant material (placed in 2 mL polypropylene tubes) were put on ice, and one minute later supplemented with 700 µL of cold (4 °C) phenol extraction buffer, containing 0.7 mol/L sucrose, 0.1 mol/L KCl, 5 mmol/L ethylenediaminetetraacetic acid (EDTA), 2% (*v*/*v*) mercaptoethanol, and 1 mmol/L phenylmethylsulfonyl fluoride (PMSF) in 0.5 mol/L tris-HCl buffer (pH 7.5). After vortexing for 30 s, 700 µL of cold phenol (4 °C) saturated with 0.5 mol/L tris-HCl buffer (pH 7.5) were added. Samples were vortexed for 30 s, shaken for 30 min at 900 rpm (4 °C), and centrifuged at 5000 g for 30 min (4 °C). The upper phenolic phase was collected in new 1.5 mL polypropylene tubes, and washed two times with equal volumes of the phenol extraction buffer (vortexing 30 s, shaking for 30 min at 900 rpm, at 4 °C and centrifugation at 5000 *g* for 15 min at 4 °C). The proteins were precipitated by adding a five-fold volume of ice-cold methanolic 0.1 mol/L ammonium acetate overnight at −20 °C. The next morning, the samples were centrifuged (10 min, 5000 g, 4 °C), and the supernatants were discarded. The pellets were washed twice by re-suspending in two volumes (relative to the volume of the phenol phase) of ice-cold methanol, and twice with the same volume of ice-cold acetone. After re-suspending, the samples were centrifuged (5000 *g*, 10 min, 4° C). The pellets were dried under air flow, reconstituted in 100 µL of shotgun buffer (8 mol/L urea, 2 mol/L thiourea, 0.15% AALS in 100 mmol/L tris-HCl, pH 7.5), and protein contents were determined by Bradford assay performed in a 96-well plate format according to Schmidt and co-workers [63]. The results of the assay were validated by SDS-PAGE as described by Greifenhagen and co-workers [64].

### 4.5. Tryptic Digestion

The tryptic digestion was performed as described by Frolov and co-workers [65] with minor modifications. In detail, the 70 µg aliquots of protein were supplemented with the shotgun buffer (see previous section) and 10 µL of 50 mmol/L TCEP (prepared in the shotgun buffer without AALS) to obtain a total volume of 100 µL. Disulfides were reduced during 30 min at 37 °C under continuous shaking (450 rpm). After cooling the samples to room temperature (RT), the proteins were alkylated with iodoacetamide (11 µL, 0.1 mol/L in 50 mmol/L aq. NH4HCO3) during 60 min at 4 °C in darkness. Afterwards, the samples were diluted with 875 µL of 50 mmol/L ammonium bicarbonate, and trypsin (0.5 g/L in 50 mmol/L aq. NH4HCO3) was added twice at the enzyme/protein ratio of 1:20 and 1:50. The proteins were hydrolyzed at 37 °C under continuous shaking (450 rpm) for 5 and 12 h, respectively. The completeness of tryptic digestion was verified by SDS-PAGE (as described above), and AALS was destroyed by addition of 103 µL of 10% (*v*/*v*) trifluoroacetic acid (TFA, final concentration 1% *v*/*v*) and incubation for 20 min at 37 °C under continuous shaking (450 rpm). Afterwards, the digests were desalted by solid phase extraction (SPE) using in-house prepared stage-tips (i.e., polypropylene pipette tips (200 µL) filled with six layers of C18 reversed phase material (Empower™ SPE discs)) [34]. The eluents were driven by centrifugal force (2500× *g*) after placing stage-tips in 2 mL polypropylene tubes using appropriate adaptors. The stage-tips were conditioned with 200 µL of methanol, equilibrated with 200 µL 0.1% (*v*/*v*) trifluoroacetic acid (TFA), before samples were loaded and washed with two 200 µL-portions of 0.1% (*v*/*v*) formic aid (FA). Afterwards, stage-tips with adaptors were transferred in new polypropylene tubes, and retained peptides were sequentially eluted with 40%, 60%, and 80% (*v*/*v*) acetonitrile in aqueous (aq.) 0.1% (*v*/*v*) FA, as proposed by Spiller and co-workers [66]. The resulting eluates were freeze-dried overnight under reduced pressure in a CentriVap Vacuum Concentrator (Labconco, Kansas City, MO, USA), and stored at −20 °C before analysis. 

### 4.6. LC-MS Experiments

Individual tryptic digests (500 ng, 10 µL) dissolved in 3% (*v*/*v*) acetonitrile in 0.1% (*v*/*v*) aq. TFA were loaded onto an Acclaim PepMap 100C18trap column (300 µm × 5 mm, 3 µm particle size) during 15 min at the flow rate of 30 µL/min. Peptides were separated at the flow rate of 300 nL/min on an Acclaim PepMap 100C18 column (75 µm × 250 mm, particle size 2 µm) using an Ultimate 3000 RSLC nano-HPLC system coupled on-line to an Orbitrap Fusion Tribrid mass spectrometer via a nano-ESI source equipped with a 30 µm ID, 40 mm long steel emitter (all Thermo Fisher Scientific, Bremen, Germany). The eluents A and B were 0.1% (*v*/*v*) aq. FA and 0.08% (*v*/*v*) aq. FA in acetonitrile, respectively. The peptides were eluted with linear gradients ramping from 1% to 35% B over 90 min, followed by 35% to 85% eluent B over 5 min. The column was washed for 5 min, and re-equilibrated at 1% eluent B for 10 min. The nano-LC-Orbitrap-MS analysis relied on data-dependent acquisition (DDA) experiments performed in the positive ion mode, comprising a survey Orbitrap-MS scan and MS/MS scans for the most abundant signals in the following 5 s (at certain tR), with charge states ranging from 2 to 6. The mass spectrometer settings and DDA parameters are summarized in the Appendix A. The mass spectrometry proteomics data have been deposited to the ProteomeXchange Consortium via the PRIDE partner repository [67] with the dataset identifier PXD013479.

### 4.7. Data Analysis and Post-Processing

Identification of peptides, annotation and label-free quantification of proteins relied on the Progenesis QIP software (Waters GmbH, Eschborn, Germany). After peak alignment, spectral and peptide filters were applied (Appendix A), and thereby selected MS/MS spectra were searched with SEQUEST engine against a combined database, containing protein sequences of three legume species closely related to pea (Appendix A), as was recently proposed by Matamoros and co-workers [44]. The database search settings are summarized in Appendix A. Afterwards, the resulted pepXML file (obtained after the search against decoy database, FDR < 0.05) was imported to Progenesis QIP software for relative quantification of identified differentially expressed proteins based on Hi-N algorithm, picking the three most abundant peptides for quantification. Finally, the proteins meeting the filter criteria (listed in Appendix A), were exported for statistical interpretation in Perseus 1.6.0.0 software (Max-Planck Institute of Biochemistry, Martinsried, Germany) [68]. This included logarithmical (log2) transformation of analyte abundances, and normalization by uniting vectors in individual experiments (columns) and by z-score based on median calculated for individual proteins (rows). Hierarchical clustering relied on Pearson correlation coefficient to cluster individual experiments and Spearman correlation coefficient to cluster individual proteins. After subsequent manual adjustment of the heat-map-based clusters, specific profile plots were built for each of them. Annotation of individual proteins relied on the original sequence database. For the proteins, annotated as “uncharacterized”, further information was derived from the Uniprot database [69]. 

For qualitative characterization of seed proteome, the raw files were directly analyzed in Proteome Discoverer 2.2 using the search parameters described above (Appendix A). Venn diagrams were built by means of the InteractiVenn tool [70]. Thereby, only the proteins and protein groups (i.e., non-redundant proteins) identified with at least one unique peptide were considered. For building Venn diagrams, all specifically modified peptides were considered as unique species. 

Functional annotation of the identified proteins relied on the Mercator 4 (v1.0) web application [35]. The results were interpreted and visualized by custom Rscripts (v 3.4.4). The closest homologues of the analyzed proteins to the Arabidopsis proteins were identified with reciprocal best-hit methods. Prediction of intracellular localization relied on the SUBA4 tool [43]. 

## 5. Conclusions

Bottom-up shotgun proteomics is a powerful tool in legume seed research. Here we successfully applied it to probe seed metabolic differences related to simultaneous inoculation of low- and high-EIBSM pea plants with rhizobia and AM fungi (i.e., the conditions mimicking the real plant rhizosphere). Thus, the high-EIBSM genotype responded to the inoculation with a prolongation of the seed filling period. This effect was due to changes in expression of the proteins involved in central energy metabolism and protein biosynthesis and folding. Of course the presented data are preliminary, and in the future these proteomics studies might be complemented with other methods of functional genomics—metabolomics and transcriptomics (i.e., a multi-omics approach can be employed). Besides, genome-wide association studies (GWAS) might help to discover novel determinants of the beneficial traits. It is important to mention; however, that efficiency of data interpretation and integration of different approaches will dramatically increase when the sequencing of the pea genome is accomplished.

## Figures and Tables

**Figure 1 molecules-24-01603-f001:**
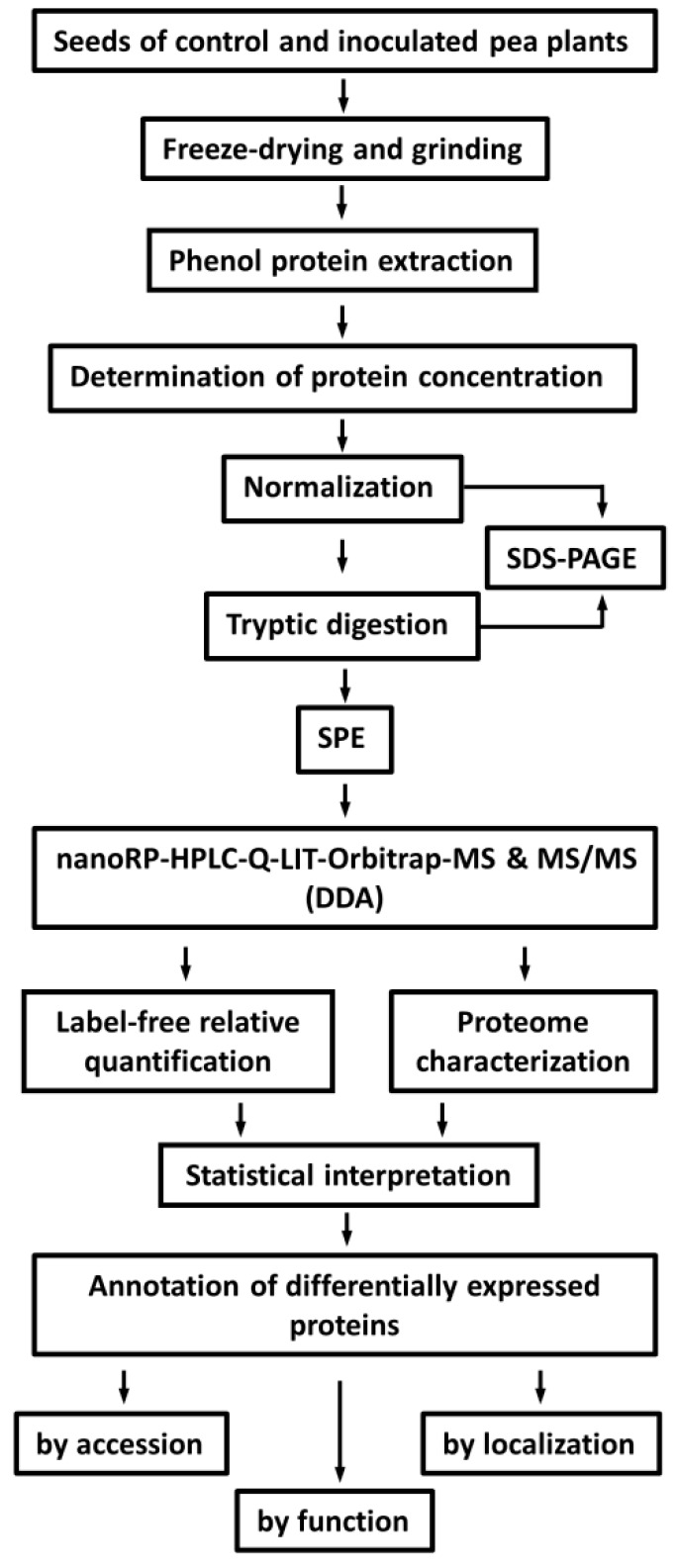
Experimental workflow for the analysis of a pea seed proteome.

**Figure 2 molecules-24-01603-f002:**
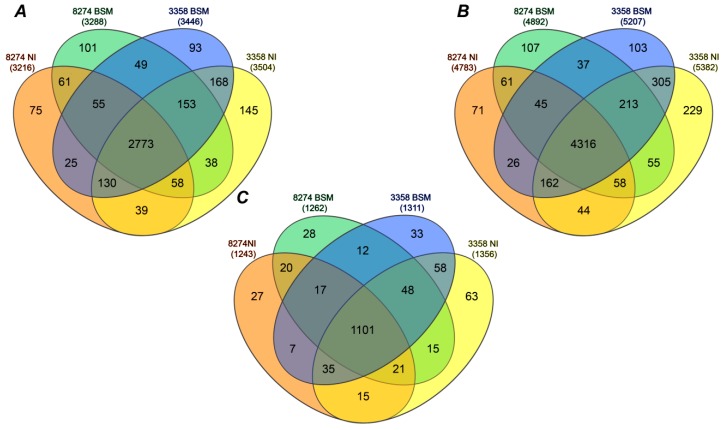
The numbers of tryptic peptides (**A**), possible proteins (**B**), and non-redundant proteins (protein groups, **C**) identified in seeds of pea (*P. sativum* L) plants, lines K-8274 (high efficiency of interaction with soil microorganisms (EIBSM), **A**) and K-3358 (low EIBSM, **B**), grown with (BSM, beneficial soil microorganisms) and without (NI, not inoculated) simultaneous colonization of pea roots with rhizobia and arbuscular mycorrhizae (AM) fungi *R. irregularis*. The pea seed protein tryptic digests (*n* = 3) were analyzed by nano-high performance liquid chromatography-electrospray ionization mass spectrometry (nanoHPLC-ESI-Q-Orbitrap-MS) in DDA mode.

**Figure 3 molecules-24-01603-f003:**
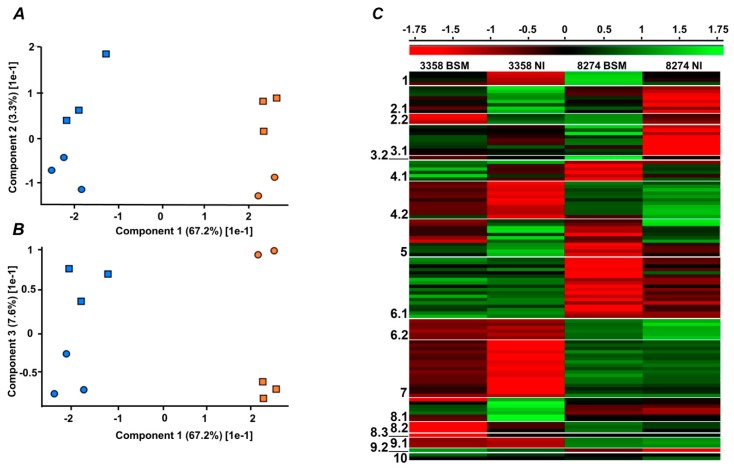
Post-processing of the label-free quantification data, acquired in nanoHPLC-ESI-Q-Orbitrap-MS/data-dependent acquisition experiments, performed with seed protein tryptic digests of pea (*P. sativum* L) plants, lines K-8274 (high EIBSM, **A**) and K-3358 (low EIBSM, **B**), grown with (BSM, beneficial soil microorganisms) and without (NI, not inoculated) simultaneous colonization of pea roots with rhizobia and arbuscular mycorrhizae (AM). The K-8274 (orange) and K-3358 (blue) pea lines could be separated by the first component (**A**,**B**), whereas BSM (squares) and NI (circles) were separated by the second (**A**) and third (**B**) components. Hierarchical clustering was done for average group values, calculated by three biological replicates (**C**). Post-processing relied on Perseus software (*n* = 3). For the original Perseus export data (i.e., prior manual verification of clusters) see Appendix A.

**Figure 4 molecules-24-01603-f004:**
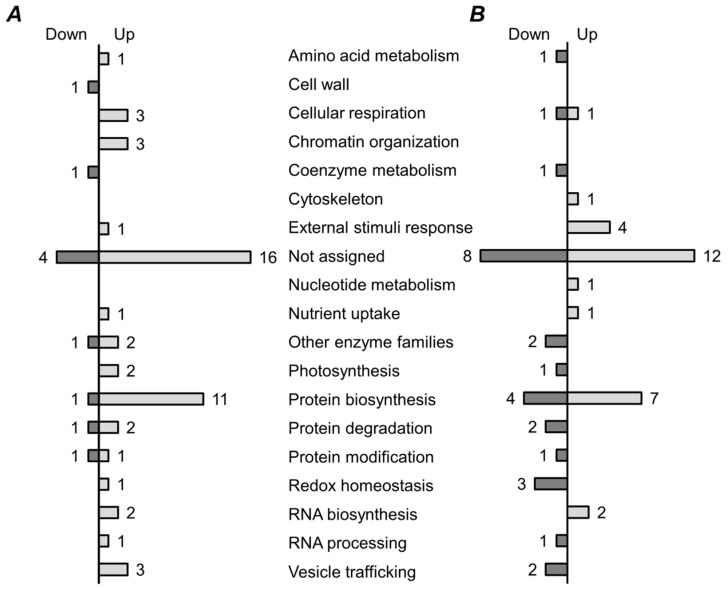
Functional annotation of proteins, differentially regulated in seeds of pea (*P. sativum* L) lines K-8274 (**A**) and K-3358 (**B**), characterized with high and low EIBSM, respectively. Functional annotation relied on Mercator tool [35].

**Figure 5 molecules-24-01603-f005:**
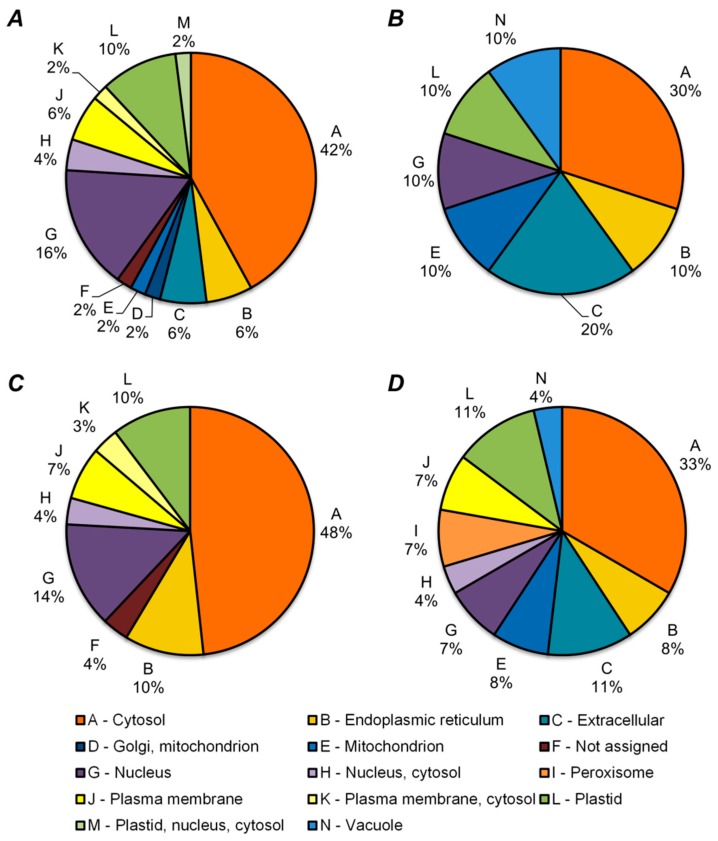
Sub-cellular localization annotation of differentially expressed proteins of seeds of pea (*P. sativum* L) as accessed in Progenesis QIP (ANOVA, *p* ≤ 0.05). ***A***—proteins up-regulated within line K-8274; ***B***—proteins down-regulated within line K-8274; ***C***—proteins up-regulated within line K-3358; ***D***—proteins down-regulated within line K-3358. Prediction of the cellular localization relied on SUBA4 tool [43].

**Table 1 molecules-24-01603-t001:** Differentially expressed pea proteins, identified in the seeds of *P. sativum* lines K-8274 and K-3385, characterized with a high and low EIBSM, respectively, and grown in presence and absence of a complex symbiosis with *Rhizobium leguminosarum* bv. *viciae* (strain RCAM 1026) and *R. irregularis* strains BEG144, BEG53, and S7.

Clusters of Protein ^a^ (8274/3358)	Nr.	Proteins	log_2_ Fold Change ^e^	Anova*p* ^f^	*q* ^g^
Accession	Description ^b^	Function ^d^	8274	3358	BSM	NI
**1. Up/Up**
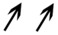	1^+^	Q1S053	Probable histone H2A.3	Chromatin organization	1.5	NS	NS	1.3	0.012	0.032
	2^+^	I3SCW0	Uncharacterized; HSP20-like chaperone ^c^	External stimuli response	2.7	1.7	NS	NS	0.023	0.043
	3	A0A072UBI6	Small hydrophilic plant seed protein	Not assigned	1.7	1.2	2.3	1.6	0.016	0.038
4	Lj0g3v0065729.1	Uncharacterized; 60S ribosomal protein L35-like ^c^	Protein biosynthesis	1.1	0.6	NS	1.6	0.011	0.029
**2. Up/Down**
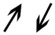	**2.1**	5^+^	Medtr1g102870.1	1,2-dihydroxy-3-keto-5-methylthiopentene dioxygenase	Amino acid metabolism	1.9	−1.3	NS	NS	0.011	0.032
6	A0A072VC98	ATPase. AAA-type. CDC48 protein	Not assigned	1.0	−0.6	3.5	3.2	0.024	0.044
7	I3T832	Uncharacterized; response to oxidative stress, heme binding, peroxidase activity ^c^	1.6	−0.7	−0.8	1.0	0.005	0.018
8	Lj6g3v1880130.1	ATP-dependent (S)-NAD(P)H-hydrate dehydratase	1.3	−1.0	−0.6	1.3	0.023	0.044
9	A0A072U0B5	UDP-glucosyltransferase family protein	Other enzyme families	0.9	−0.8	2.0	1.9	0.012	0.031
10	G7IEE7	Xyloglucanase-specific endoglucanase inhibitor p.	Protein degradation	1.0	NS	−4.6	−4.5	0.002	0.014
11^+^	B7FH22	Uncharacterized; oxidoreductase activity ^c^	Redox homeostasis	2.3	−0.7	NS	0.8	0.024	0.043
12^+^	G7L8T3	Guanosine nucleotide diphosphate dissociation inhibitor	Vesicle trafficking	1.7	−1.7	−5.1	−5.2	0.020	0.043
	**2.2**	13	G7IJ13	Proteasome subunit alpha type	Protein degradation	1.3	−0.7	−2.3	−2.1	0.025	0.044
14	B7FLD1	Putative uncharacterized; Nop domain superfamily (pre-RNA processing ribonucleoproteins) ^c^	RNA processing	1.8	−1.8	0.6	NS	0.001	0.013
15	A0A072UGB7	Clathrin heavy chain	Vesicle trafficking	1.4	−1.4	3.7	2.6	0.034	0.050
**3. Up/Steady**
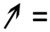	**3.1**	16	A0A072W1H5	ATP synthase subunit beta	Cellular respiration	3.0	NS	NS	2.7	0.003	0.017
17	I3SN66	Uncharacterized; triose-phosphate isomerase activity, chloroplast organization ^c^	2.7	1.0	−0.6	−0.8	0.004	0.017
18	G7IUE0	LRR receptor-like kinase family protein	Not assigned	2.1	NS	NS	NS	0.010	0.029
19	G7J538	GDP-fucose protein O-fucosyltransferase	1.6	NS	−0.7	−0.9	0.023	0.044
20	V7CPQ1	Uncharacterized; ATP binding ^c^	1.8	NS	0.8	2.9	0.002	0.015
21^+^	A2Q582	Aldo/keto reductase	Other enzyme families	1.9	NS	NS	3.3	0.016	0.038
22	A0A072UJ10	Cytoplasmic ribosomal protein S13	Protein biosynthesis	3.9	0.8	2.6	2.5	0.002	0.015
23	I3T617	60S ribosomal L35-like protein	4.0	1.4	NS	2.4	0.005	0.018
24^+^	Q5QQ34	Coatomer epsilon subunit	Vesicle trafficking	2.8	NS	NS	1.5	0.019	0.041
	**3.2**	25	I3SHC8	Uncharacterized; ribosome biogenesis ^c^	Protein biosynthesis	1.7	NS	NS	−0.6	0.008	0.025
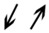	**4.1**	26	V7BQ03	Uncharacterized; carboxy-lyase activity, magnesium ion binding, thiamine pyrophosphate binding ^c^	Carbohydrate metabolism	NS	NS	−2.9	0.8	0.037	0.050
27	G7KBA2	17.6 kDa class I heat shock protein	External stimuli response	NS	1.0	−1.2	−1.3	0.003	0.015
28	A0A072TF91	Heat shock protein HSP20. putative (Fragment)	Not assigned	NS	NS	2.5	1.8	0.030	0.046
29	Lj3g3v0324640.1	Lipoxygenase	Other enzyme families	NS	NS	−1.7	1.1	0.004	0.018
30^+^	V7BZK0	Lipoxygenase	NS	NS	2.1	2.9	0.004	0.019
	**4.2**	31	G7L0I7	Cobalamin-independent methionine synthase	Amino acid metabolism	NS	NS	−1.1	−5.0	0.012	0.031
	32	G7L831	TCP-1/cpn60 chaperonin family protein	Cytoskeleton	NS	0.9	−0.7	−1.7	0.023	0.044
	33	G7JSC7	NB-ARC domain disease resistance protein	Not assigned	0.6	0.6	−1.2	−1.3	0.002	0.014
	34	Lj1g3v0411500.1	Uncharacterized; Myb/SANT-like domain (nuclear DNA-binding proteins, nuclear receptor co-repressors) ^c^	NS	0.6	NS	1.8	0.016	0.038
	35	V7ARA2	Uncharacterized; Ca^2+^ binding ^c^	NS	NS	1.6	2.1	0.001	0.012
	36	V7BSM8	Annexin	NS	NS	0.6	−0.8	0.001	0.013
	37	V7AR99	Uncharacterized; lipase activity ^c^	NS	1.0	1.7	2.6	0.016	0.038
	38	B7FIG5	Putative uncharacterized; oxidoreductase activity, acting on the CH–CH group of donors ^c^	Nucleotide metabolism	NS	0.7	1.5	2.6	0.001	0.013
	39	A0A072V122	tRNA-binding region domain protein	Protein biosynthesis	NS	NS	2.4	1.9	0.024	0.044
	40	V7AUC2	Uncharacterized; RNA 3′-end processing, RNA polyadenylation ^c^	RNA processing	NS	NS	−0.6	NS	0.013	0.032
	41^+^	I3SYE6	1,2-dihydroxy-3-keto-5-methylthiopentene dioxygenase	Amino acid metabolism	NS	NS	NS	-0.6	0.009	0.027
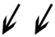	42	G7J834	Glucose-1-phosphate adenylyltransferase	Carbohydrate metabolism	NS	NS	−0.9	−0.6	0.024	0.044
43	G7JM88	Lethal leaf-spot protein. putative	Coenzyme metabolism	NS	−1.1	−0.6	−1.1	0.018	0.040
44	Lj4g3v2371610.1	Probable glycine cleavage T-protein family (aminomethyl transferase)	−0.9	NS	−2.1	1.5	0.004	0.017
45^+^	Medtr5g019780.1	Cupin family protein	Not assigned	NS	−1.0	NS	1.1	0.013	0.033
46	A0A072TDJ4	TCP-1/cpn60 chaperonin family protein (Fragment)	−1.4	−1.1	1.4	1.2	0.037	0.051
47	G7IDU4	Protein disulfide isomerase-like protein	−1.1	−1.0	−1.6	−1.4	0.003	0.017
48	V7AJE4	Probable defense response, ADP binding	NS	−2.0	1.8	1.7	0.024	0.044
49^+^	B7FJF0	Xylose isomerase	Other enzyme families	−2.7	−3.5	NS	1.6	0.021	0.043
50^+^	G7ILF2	60S ribosomal protein L26-1	Protein biosynthesis	−0.6	NS	−1.0	−1.2	0.031	0.047
51^+^	I3T560	Superoxide dismutase	Redox homeostasis	NS	−0.8	−1.4	1.7	0.023	0.043
**6. Down/Steady**
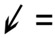	**6.1**	52^+^	I3SU69	Uncharacterized; argininosuccinatelyase activity ^c^	Amino acid metabolism	NS	NS	−0.9	−1.5	0.001	0.010
53	G7J530	Argininosuccinatelyase	NS	NS	−0.6	−0.8	0.001	0.009
54^+^	A0A072TSD1	Pectin acetylesterase	Cell wall	−0.9	NS	1.3	1.2	0.008	0.026
55	G7JFK1	Heat shock 70 kDa protein	External stimuli response	NS	0.8	−0.7	−0.7	0.001	0.010
56^+^	V7C9P5	Uncharacterized; ATP binding ^c^	NS	0.8	3.5	3.2	0.001	0.010
57	A0A072U2T6	Translin-like protein	Not assigned	NS	NS	0.9	0.9	0.000	0.008
58^+^	A0A072TMR3	Late embryogenesis abundant protein	−1.3	NS	1.2	0.7	0.000	0.001
59^+^	B1NY79	Cold-acclimation specific protein 15	NS	0.9	−0.7	1.3	0.005	0.020
60^+^	B5U8K3	Convicilin storage protein 1	NS	NS	NS	1.2	0.001	0.010
61^+^	I3S2D8	Uncharacterized; Mitochondrial inner membrane translocase subunit ^c^	2.4	NS	−0.6	−1.4	0.000	0.000
62^+^	V7BVA1	Uncharacterized; QWRF domain family, microtubule-associated ^c^	−0.6	−0.6	−1.3	NS	0.000	0.000
63	I3T8A0	Glutamine synthetase	Nutrient uptake	NS	NS	NS	1.4	0.025	0.044
64	G7IS29	Lipoxygenase	Other enzyme families	NS	NS	NS	−0.8	0.024	0.044
65^+^	Medtr1g094155.1	Probable serine carboxypeptidase-like protein	Protein degradation	−1.0	−0.6	NS	−0.6	0.023	0.043
66	G7I549	26S proteasome non-ATPase regulatory subunit-like protein	NS	NS	0.7	0.6	0.034	0.050
67	A0A072TQN5	Phosphatase 2C family protein	Protein modification	NS	NS	0.9	0.6	0.003	0.016
68	B7FMC4	Putative uncharacterized; Glutathione S-transferases terminal domain ^c^	−0.6	NS	1.6	NS	0.023	0.044
69	G7LH03	Glycosyltransferase	Secondary metabolism	NS	NS	−2.1	−2.7	0.000	0.008
	**6.2**	70	G7JPY4	Delta-1-pyrroline-5-carboxylate dehydrogenase	Amino acid metabolism	NS	NS	−0.9	−0.7	0.004	0.017
	71	A0A072VNG1	Uncharacterized protein	Not assigned	NS	NS	NS	2.1	0.002	0.014
	72	Lj1g3v3690420.1	Elongation factor 1-alpha	Protein biosynthesis	NS	NS	NS	2.7	0.026	0.044
	73	G7IVL9	U-box kinase family protein	Protein modification	NS	NS	NS	−0.8	0.034	0.050
	74	Lj0g3v0348019.1	Transcription factor	RNA biosynthesis	NS	NS	0.8	2.9	0.002	0.014
	75	Q93XA4	Homeodomain leucine zipper protein HDZ2	NS	NS	−1.1	NS	0.007	0.022
**7. Steady/Up**
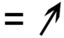	76	V7AU77	Uncharacterized; lactoylglutathionelyase activity ^c^	Cellular respiration	0.8	NS	−0.9	NS	0.030	0.046
	77	G7IHB8	Core histone H2A/H2B/H3/H4	Chromatin organization	0.8	NS	−0.7	−0.7	0.025	0.044
	78^+^	Q38JC8	Temperature-induced lipocalin	Not assigned	1.0	1.7	1.1	0.6	0.015	0.036
79	G7JPM2	Uro-adherence factor A. putative	NS	0.9	−1.0	−1.0	0.027	0.046
80	V7ALP7	Annexin	0.6	0.6	1.7	1.9	0.013	0.032
81	B7FJY0	Annexin	0.8	0.6	−0.9	3.6	0.010	0.029
82	V7B712	Hexosyltransferase	0.9	1.0	−0.9	NS	0.035	0.050
83	V7BYE1	Uncharacterized (Fragment); Leucine-rich repeat domain superfamily ^c^	0.9	1.4	−1.4	0.6	0.037	0.050
	84^+^	A0A072U7T5	F-box/RNI/F box domain-like domain protein	0.7	4.6	6.9	5.8	0.003	0.017
	85	B7FK47	Ferritin	Nutrient uptake	0.7	0.9	2.4	3.3	0.031	0.048
	86	A0A072UUP4	60S ribosomal protein L18a	Protein biosynthesis	NS	0.9	1.4	1.1	0.029	0.046
	87	A0A072VAP6	60S ribosomal protein L17A	0.6	NS	8.9	8.9	0.032	0.049
	88	A0A072VJE7	40S ribosomal protein S5-2	NS	0.9	0.7	2.4	0.028	0.046
	89^+^	G7IH13	Translation elongation factor EF-2 subunit	0.6	1.7	−1.7	−1.8	0.025	0.043
	90	B7FMQ6	60S ribosomal L23-like protein	0.7	0.9	0.7	0.6	0.036	0.050
	91	V7B0F4	Uncharacterized; RNA-binding, RNA-mediated gene silencing ^c^	RNA biosynthesis	0.8	1.0	−1.8	−0.8	0.006	0.021
	92^+^	I3SRR2	Uncharacterized; transcription factor activity, sequence-specific DNA binding, zinc ion binding ^c^	0.7	0.9	−0.6	−0.8	0.011	0.032
	93	G7J2R6	110 kDa 4SNc-tudor domain protein	RNA processing	NS	NS	−1.1	−0.8	0.008	0.025
**8. Steady/Down**
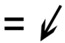	**8.1**	94	V7AWC5	4-alpha-glucanotransferase	Carbohydrate metabolism	NS	NS	−1.4	NS	0.033	0.049
	95	G9JLT6	ATP synthase subunit alpha	Cellular respiration	NS	−0.8	−0.7	−0.9	0.004	0.017
	96^+^	G7KG34	Glutamine synthetase	Nutrient uptake	NS	NS	−0.7	−1.2	0.003	0.016
	97	G7IH71	Phosphoenolpyruvate carboxylase	Photosynthesis	NS	−0.6	−1.2	−1.6	0.000	0.008
98	G7IBY1	60S ribosomal protein L3B	Protein biosynthesis	0.7	−1.3	−1.3	−1.2	0.003	0.017
99^+^	B7FN14	Uncharacterized; Involved in translation, rRNA-binding ^c^	0.7	−3.8	1.0	0.8	0.002	0.014
100^+^	A0A072TQ47	Phosphatase 2C family protein	Protein modification	0.6	NS	NS	0.8	0.015	0.036
101^+^	A0PG70	Catalase	Redox homeostasis	NS	−1.4	NS	2.7	0.021	0.043
	**8.2**	102	A0A072UKG0	Histone H2B	Chromatin organization	0.8	NS	11.4	13.5	0.005	0.018
	103	V7B8C8	Uncharacterized; translation, structural constituent of ribosome ^c^	Protein biosynthesis	0.8	−1.1	−2.1	NS	0.009	0.026
	104^+^	A0A072VE37	UTP-glucose-1-phosphate uridylyltransferase	1.5	−2.1	NS	2.2	0.008	0.026
	**8.3**	105	A0A072VJU4	Glutathione S-transferase. amino-terminal domain protein	Protein modification	NS	−1.4	0.7	1.9	0.005	0.018
**9. Steady/Steady**
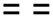	**9.1**	106	A2Q4V2	Leucine-rich repeat. plant specific	Not assigned	NS	NS	−0.9	2.5	0.000	0.008
107	I3SIG9	Chlorophyll a-b binding protein. chloroplastic	Photosynthesis	0.6	NS	NS	−0.9	0.000	0.000
108	B6DXD7	Vacuolar H+-translocating inorganic pyrophosphatase	Solute transport	NS	NS	0.9	2.1	0.008	0.025
	**9.2**	109^+^	G7IMZ3	ABA-responsive protein	Not assigned	1.4	NS	−1.6	−1.9	0.003	0.017
**10. None ^h^**
	110	A0A072TYG8	β-hexosaminidase	Protein modification	-	NS	-	-	0.000	0.000
	111	Lj0g3v0098069.1	Uncharacterized; l-type lectin-domain containing receptor kinase S.7-like ^c^	Not assigned	NS	-	-	-	0.000	0.000

Plants were grown under non-controlled light and temperature conditions in a greenhouse, as described in Materials and Methods section. The plants were harvested at the stage of mature seeds (three months after planting). The total seed protein fraction was isolated by phenol extraction, the proteins were digested by trypsin and resulted digests were analyzed by nanoHPLC-Q-Orbitrap-LIT-MS. Abbreviations: Nr.—number of protein; UDP—uridine diphosphate; LRR—leucine-rich repeat; GDP—guanidine diphosphate; NB-ARC—nucleotide-binding adaptor shared by APAF-1, R proteins, and CED-4; ADP –adenosine diphosphate. ^a^ Initial grouping of proteins by expression profiles relied on hierarchical clustering (using Spearman correlation as a distance measure) with subsequent manual correction of individual protein plots in Perseus software (if necessary); individual expression profiles were defined based on the direction of changes in protein abundance in response to inoculation with BSM; ^b^ the descriptions for individual proteins were taken from headers of corresponding fasta files; ^c^ for the proteins, annotated as “Uncharacterized” or “Putative uncharacterized”, additional information from UniprotKB was collected; ^d^ functional annotation relied on the Mercator software; ^e^ binary logarithm of fold changes (log_2_FCs) within the lines K-8274 and K-3358 is calculated for the abundance ratios BSM_K-8274_/NI_K-8274_ and BSM_K-3385_/NI_K-3385_, whereas the comparisons of the lines relied on the ratios BSM_K-3385_/BSM_K-8274_ and NI_K-3385_/NI_K-8274_; ^f^
*p* values were obtained by one-way ANOVA using Progenesis QI software; ^g^ q values were obtained with Progenesis QI software; ^h^ the tenth profile corresponds to proteins which were not found in one of the lines: A0A072TYG8 was identified and quantified only in line 3358 and Lj0g3v0098069.1 only in line 8274; **+** indicates the proteins identified in the search against a redundant sequence database and manually checked for quality of identification. NS—“Non-significant” denotes fold changes <1.5 in absolute scale or <0.6 and >−0.6 in log_2_ scale.

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
