# Peer review of "Profiling of Seed Proteome in Pea (Pisum sativum L.) Lines Characterized with High and Low Responsivity to Combined Inoculation with Nodule Bacteria and Arbuscular Mycorrhizal Fungi"

_molecules, 2019, doi:10.3390/molecules24081603_

Round 1

Reviewer 1 Report

The received publication is original and interesting as well as provides a new and important data. The experiments were performed by the group which have a large experience in plant proteomics. The authors have developed adequate experiments and the results are clear cut enough to draw convincing conclusions. Summary and introduction are adequate. The aim of the study is clearly described. The publication is generally well written. However I have a one major issue to solve. The experiment lacks of a data validation using other proteomic approaches. I am strongly suggest to perform western blot validation of observed results at least for the proteins that were expressed only in one of the groups (e.g. ß-hexosaminidase), but if possible also for the additional selected proteins from major protein groups.

Author Response

We thank the reviewer for the thoughtful review and highly appreciate the valuable comments and suggestions to improve the manuscript. Following these advices we performed all required changes in corresponding sections, as indicated in the following rebuttal addressing each aspect.

Reviewer: 1

Major remarks

Major remark 1: However I have a one major issue to solve. The experiment lacks of a data validation using other proteomic approaches. I am strongly suggest to perform western blot validation of observed results at least for the proteins that were expressed only in one of the groups (e.g. ß-hexosaminidase), but if possible also for the additional selected proteins from major protein groups.”

Answer: We understand the consideration of the Reviewer. Moreover, if our quantification would rely on spectral counting, I would, for sure, request exactly the same. However, our quantification relied on peak area-based approach. It means, the values compared between groups were derived from peak areas at extracted ion chromatograms. For IT-Orbitrap-MS, limits of detection lie in low femtomol range or even lower (and the data were acquired with the quite new highly-sensitive tribrid instrument). The linearity and precision of our approach is already validated (Frolov et al. 2014. Anal Bioanal Chem, 406(24): 5755-63) and, hence, does not need any cross-validation any more. Moreover, cross-validation with a less sensitive method would not be so helpful. We added a corresponding sentence to the Discussion:

“Recently, we validated our nanoHPLC-ion trap (IT)-Orbitrap-MS-based approach for label-free quantification and confirmed its high reliability, precision and sensitivity [31]. In our earlier studies it proved to be well compatible with other functional genomics techniques [24].” (lines 318-321).

To address the issue pointed by the reviewer, we provide graphical data, supporting our results in Supplementary information (Figure S1-5) and a corresponding remark in the text:

“(as could be proved by verification of peak integration, Figure S1-5)” (lines 175-176).

Reviewer 2 Report

In their article mamontova and colleagues describe the alterations in proteomic profiles of seeds from Pisum sativum lines characterized by high and low efficiency of interaction with beneficial soil microorganisms.

Based on the results of their studies, the authors state that proteomics is a good tool to investigate metabolic differences in pea seeds in conditions that mimick the real plant rhizosphere. Science is good and the message is of interest,  representing the novelty of the research. Obviously, as also admitted by the authors themselves, these data are preliminary and their true meaning could be understood in its entirety only after having achieved the pea genome sequence.

Author Response

We thank the reviewer for the thoughtful review and highly appreciate the valuable comments and suggestions to improve the manuscript. Following these advices we performed all required changes in corresponding sections, as indicated in the following rebuttal addressing each aspect.

Reviewer: 2

Major remarks

Major remark 1: Based on the results of their studies, the authors state that proteomics is a good tool to investigate metabolic differences in pea seeds in conditions that mimick the real plant rhizosphere. Science is good and the message is of interest,  representing the novelty of the research. Obviously, as also admitted by the authors themselves, these data are preliminary and their true meaning could be understood in its entirety only after having achieved the pea genome sequence

Answer: We agree with the reviewer and modified the conclusion section accordingly:

“Of course, the presented data are preliminary, and in the future, these proteomics studies…” (lines 594 - 595).

Reviewer 3 Report

The authors have responded adequately to the points raised in the first review. Although a lot more could have been removed to make the paper more accessible to non-specialist readers, this is a question of style and suitability for this journal. Only minor grammatical errors remain.

Round 2

Reviewer 1 Report

Dear Authors,

Thank you for your response, that convinced me to accept authors arguments and recomend an article for publication in its present form.